# Factors Associated with Primary Liver Cancer Survival in a Southern Italian Setting in a Changing Epidemiological Scenario

**DOI:** 10.3390/cancers16112046

**Published:** 2024-05-28

**Authors:** Sergio Mazzola, Martina Vittorietti, Santo Fruscione, Daniele Domenico De Bella, Alessandra Savatteri, Miriam Belluzzo, Daniela Ginevra, Alice Gioia, Davide Costanza, Maria Domenica Castellone, Claudio Costantino, Maurizio Zarcone, Barbara Ravazzolo, Giorgio Graziano, Rita Mannino, Rosalba Amodio, Vito Di Marco, Francesco Vitale, Walter Mazzucco

**Affiliations:** 1Clinical Epidemiology and Cancer Registry Unit, Azienda Ospedaliera Universitaria Policlinico di Palermo, 90127 Palermo, Italy; sergio.mazzola@policlinico.pa.it (S.M.); claudio.costantino01@unipa.it (C.C.); maurizio.zarcone@policlinico.pa.it (M.Z.); barbara.ravazzolo@policlinico.pa.it (B.R.); giorgio.graziano@policlinico.pa.it (G.G.); rita.mannino@policlinico.pa.it (R.M.); rosalba.amodio@policlinico.pa.it (R.A.); francesco.vitale@unipa.it (F.V.); walter.mazzucco@unipa.it (W.M.); 2Delft University of Technology, 2628 Delft, The Netherlands; m.vittorietti@tudelft.nl; 3PROMISE Department, University of Palermo, 90127 Palermo, Italy; danieledomenico.debella@unipa.it (D.D.D.B.); alessandra.savatteri@unipa.it (A.S.); miriam.belluzzo01@unipa.it (M.B.); daniela.ginevra@unipa.it (D.G.); alice.gioia@unipa.it (A.G.); davide.costanza@unipa.it (D.C.); vito.dimarco@unipa.it (V.D.M.); 4Istituto di Endocrinologia ed Oncologia Sperimentale del CNR (IEOS), 80100 Napoli, Italy; mariadomenicacastellone@gmail.com; 5College of Medicine, University of Cincinnati, Cincinnati, OH 45221, USA

**Keywords:** liver cancer, epidemiology, access to care, survival, cancer registries, health services, digital prevention

## Abstract

**Simple Summary:**

The epidemiology of primary liver cancer is changing according to effective primary prevention interventions, a strictly clinical follow up of chronic patients with hepatitis and successful drug treatments of virus-related liver diseases. A reduction in virus-related hepatocellular carcinomas cases was followed by an increase in the non-viral aetiology, including metabolic and cryptogenetic factors. We used cancer surveillance data to investigate determinants potentially associated with primary liver cancer survival in a southern Italian setting characterised by a high prevalence of hepatitis viral infections and overweight and obesity. Our study provided some new insights into the potential relationship between deprivation, geographic location, access to care and hepatocarcinoma survival. Comprehensive national and regional health plans should consider the evidence provided to ensure equitable access to prevention and care pathways. While entering the digital prevention era, interoperating cancer registries with other sources providing data on lifestyles and health determinants should be promoted.

**Abstract:**

A retrospective observational study utilising cancer incidence data from a population-based registry investigated determinants affecting primary liver cancer survival in a southern Italian region with high hepatitis viral infection rates and obesity prevalence. Among 2687 patients diagnosed between 2006 and 2019 (65.3% male), a flexible hazard-based regression model revealed factors influencing 5-year survival rates. High deprivation levels [HR = 1.41 (95%CI = 1.15–1.76); *p* < 0.001], poor access to care [HR = 1.99 (95%IC = 1.70–2.35); *p* < 0.0001], age between 65 and 75 [HR = 1.48 (95%IC = 1.09–2.01); *p* < 0.05] or >75 [HR = 2.21 (95%CI = 1.62–3.01); *p* < 0.0001] and residing in non-urban areas [HR = 1.35 (95%CI = 1.08–1.69); *p* < 0.01] were associated with poorer survival estimates. While deprivation appeared to be a risk factor for primary liver cancer patients residing within the urban area, the geographic distance from specialised treatment centres emerged as a potential determinant of lower survival estimates for residents in the non-urban areas. After balancing the groups of easy and poor access to care using a propensity score approach, poor access to care and a lower socioeconomic status resulted in potentially having a negative impact on primary liver cancer survival, particularly among urban residents. We emphasise the need to interoperate cancer registries with other data sources and to deploy innovative digital solutions to improve cancer prevention.

## 1. Introduction

Despite general improvement, primary liver cancers have poor survival outcomes worldwide, with little differences between low-resource and high-income countries explained by diagnostic and therapeutic advances [1,2,3]. Globally, liver cancer represents the third cause of cancer death, with an estimated incidence of about nine hundred thousand new cases per year [4,5]. Primary liver cancer includes hepatocellular carcinoma (HCC) (75–85% of cases) and intrahepatic cholangiocarcinoma (10–15%), as well as other less frequent morphologic types. Most of the HCC cases are associated with chronic infections by hepatitis B virus (HBV) or hepatitis C virus (HCV). However, in the last few years, the epidemiology of HCC has been continuously evolving according to effective primary prevention interventions [6], the wider use of a strictly clinical follow up of chronic patients with hepatitis and successful drug treatments of virus-related liver diseases [6]. As a consequence of the reduction in HCC virus-related cases, an increase in HCC patients’ age and a progressive expansion of non-viral liver cancer cases, such as metabolic and alcohol-related liver disease, were highlighted [7,8], therefore suggesting the need to strengthen prevention strategies against non-viral risk factors associated with liver diseases, including hepatocellular carcinoma [9]. Of interest, studies on the impact of emerging risk factors, other than the well-documented risk from alcohol consumption such as non-alcoholic fatty liver disease [10], highlighted how cancer survival was associated with the socioeconomic and demographic status of patients [11] and focused on the pathways of care leading to a cancer diagnosis [12]. More in depth, several sociodemographic characteristics have been associated with HCC, particularly in cirrhotic patients, with ageing and male gender having a predominant role [13,14]. Furthermore, an association between lower socioeconomic status and a higher incidence of HCC has been reported globally [12]. Specifically, living in the most deprived areas was associated with an approximately twofold increase in HCC incidence, whereas contrasting results have been reported on the association between deprivation and survival following a diagnosis of HCC [15]. Of interest, the association between deprivation and poor health outcomes highlighted in more deprived areas implies that disparities between rural and urban areas may be explained by different lifestyles [16], which in turn have implications for health and the use of healthcare services [17]. Socioeconomic factors do not only affect cancer risk but also access to healthcare, the possibility of a timely diagnosis as well as prognosis and therapeutic outcomes [13]. Hence, identifying risk factors and determining their influence on HCC survival can be challenging.

In Italy, high HCC incidence and mortality rates have been highlighted, with a north-to-south increasing gradient [15,18], as compared with other European and Western countries [19,20], with about 90% of primary liver neoplasm estimated to be attributable to HBV and HCV infections [21]. More recently, in Italy, a significant decrease in HCC incidence, together with an improvement in survival outcomes, was reported [22].

Starting from a previously validated model [23], we used data from a population-based cancer registry to investigate determinants associated with primary liver cancer survival, including difference in access to care and sociodemographic factors, such as deprivation and residence, in HCC patients from a southern Italian setting characterised by a high prevalence of hepatitis viral infections and an increasing population of overweight and obesity [24,25].

## 2. Materials and Methods

We conducted a retrospective observational study using the cancer surveillance data registered by the Palermo Province Cancer Registry in the period 2006–2019, obtaining a sample of 2687 individuals (age range 15–99 years) who received a diagnosis of primary malignant liver neoplasm (C22.0), behaviour code 3, in the International Classification of Diseases for Oncology–ICD-O-3, Third Edition [26,27]. Cancers arising from intrahepatic biliary ducts (C22.1) were not included. We categorised primary liver cancer patients by access to care according to a previously validated linkage algorithm that allowed us to identify patients with “easy access to care”, the ones having a higher number of contacts with healthcare service as confirmed by the linkage with at least two different healthcare sources (hospital discharge records, ambulatory discharge records and/or pathological anatomy reports) and patients having “poor access to care”, which were identified by a single data source [23]. 

To obtain a deprivation measure for every single section of the census in the Palermo province, we used the official data from the demographic and housing census for 2001 [28], and we assembled a composite indicator using (1) the percentage of the population with a low level of education, (2) percentage of the unemployed population, (3) percentage of the population living in a rented place, (4) percentage of the population living in crowded houses and (5) percentage of the population living in a single-parent family, as proposed by Caranci et al. [29]. Instead of normalising the variables, adding them together and categorising the resulting index in quintiles as previously suggested [30], after having standardised the variables, we performed a principal components analysis to obtain weights for the original variables that might better reflect the socioeconomic factors influencing each census section. Through this methodological approach, we obtained two deprivation components [29]:

(1) Deprivation component 1 (DC.1), given by the linear combination of the first 4 deprivation variables (people with a low education level; unemployed people; living in a rented place; living in crowded houses): people with a low education level × (0.541) + unemployed population × (0.517) + population living in a rented place × (0.528) + living in crowded houses × (0.401); 

(2) Deprivation component 2 (DC.2), given by the linear combination of living in single-parent households: living in a single-parent family × (−0.941).

We therefore considered them as separated deprivation indexes, and we defined a median threshold value of 0 to distinguish high deprivation (DC.1 > 0 or DC.2 > 0) from low deprivation (DC.1 < 0 or DC.2 < 0) [30,31].

We then applied a logistic regression model considering the information on cancers defined by death certificate only (DCO), considered as a proxy of the poorest access to care as compared to the other categories, as the dependent variable, while including the following covariates: weighted deprivation components DC.1 and DC.2; the municipality of residence (“urban”, corresponding to Palermo city; “not urban”, including the remaining 81 municipalities of the province); sex (male; female) and age classes (15–55; 55–65; 65–75; 75+ years old). 

Our series included patients with a follow up during the SARS-CoV-2 pandemic period (2020–2023). To assess the possible influence of the pandemic on the overall 5-year survival, we divided the patients into two subgroups according to the incidence period 2006–2014 with the highest follow up to 2019 (before the pandemic) and 2015–2018 with the highest follow up to 2023 (during the pandemic), and we then compared the two 5-year net survival curves.

Afterwards, we excluded the poorest category of the DCO and balanced the sociodemographic differences between the groups of poor and easy access using a propensity score matching procedure. To this end, we estimated the propensity scores using a probit model considering as a response variable the access to care (“poor” against “easy”) and explanatory variables “deprivation components DC.1” and “DC.2”, the “municipality of residence (urban; not urban)”, “sex (male; female)” and “age classes (15–55; 55–65; 65–75; 75+ years old)”, and we matched the observations using the nearest neighbour method [31].

Therefore, we considered two flexible hazard-based regression models with fixed effects, one using the original dataset and one using the new dataset obtained after propensity score matching, to verify whether the abovementioned covariates were associated with HCC survival and to assess the pure effect of the distance from healthcare on HCC survival.

Statistical analyses were performed using the IDE software Rstudio (version 3.4.1 of 2017-06-30) for R (version 2.1) [32]. Net survival was estimated by using the “relsuv” package [33]. The flexible hazard-based regression model was estimated by the “mexhaz” package [34]. The propensity matching model was applied using the “MatchIt” package [35]. The two-sided statistical significance of the difference between paired survival estimates was set at 0.05. 

## 3. Results

Of the 2,687 adults with a primary liver tumour (C22.0) included in this study, 1754 (65.3%) were males and 933 (34.7%) females (Table 1), with an overall mean age of 71 years old and the most frequent age at diagnosis category represented by the patients aged > 75 years old (40.1%), followed by the age groups 65–74 years old (35.5%), 55–64 years old (16.6%), 45–54 (6.2%) and 15–44 (1.6%) (Table 1).

Most of the HCC patients were residents in the urban area (62% versus 38% residing in the remaining municipalities of the non-urban area) (Table 1). 

The easy access to care group consisted of 1744 (64.9) HCC patients, while the patients with poor access to care were 700 (26.1), and the remaining 243 (9.0) ones, defined by DCO, belonged to the group with the poorest access to care (Table 1). Males were more represented in both groups of HCC patients with easy (69.6%) and poor access to care (60.0%), while females slightly prevailed (50.6%) in the group with the poorest access to care (*p*-value: <0.0001) (Table 1). 

The mean age at diagnosis of HCC patients with easy access to care (69 years old) was significantly lower as compared to the ones of patients with poor access to care (72 years old) and with the poorest access to care (79 years old) (*p*-value: <0.0001), with the ≥75 years old category being the most represented in all the age groups (about two-thirds of HCC patients with the poorest access to care belonged to this age category at diagnosis), except for HCC patients with easy access to care where the 65–74 age group prevailed (39.3%) (*p*-value: <0.00001) (Table 1). Statistically significant differences (*p*-value < 0.0001) were also highlighted in the distribution of the HCC cases according to the place of residence: HCC patients residing in the urban area were more represented than the ones from the non-urban municipalities both in the groups with easy access to care (65.5% vs 34.5%) and poor access to care (59.0% vs 41.0%), while in the group of patients with the poorest access, this proportion was inverted (51.0% vs 49.0%) (Table 1).

Table 2 presents the results of the logistic regression model fitted for estimating the probability of having the poorest access to care. 

A high level of deprivation (DC.1 > 0) [OR = 1.41 (95%CI = 1.07–1.87); *p*-value: <0.05], the female gender [OR = 1.68 (95%CI = 1.27–2.22); *p*-value: <0.0001], the age group ≥ 75 years old [OR = 5.08 (95%CI = 2.51–12.16); *p*-value: <0.0001] and living in a non-urban area [OR = 1.97 (95%CI = 1.45–2.67); *p*-value: <0.0001] were all factors associated with a higher probability of the poorest access to care.

Figure 1 shows the 5-year net survival curves of two subgroups of HCC patients, the first one including patients receiving a diagnosis in the period 2006–2014, and the second one considering patients with a diagnosis of primary liver cancer in the years 2015–2018, the latter one characterised by survival estimates intercepting the COVID-19 pandemic: no statistically significant difference in survival was found in the comparison between these subgroups (* Log Rank Test: *p*-value = 0.10).

Figure 2 shows the comparison of the mean difference values of the considered covariates between the two balanced groups of patients (“poor” versus “easy” access to care), before and after propensity score matching: after matching, all the covariates had an absolute standardised mean difference between the two groups of less than 0.05.

The results of the fixed effects risk regression model (Table 3) highlighted that a high deprivation level (DC.1 > 0) [HR = 1.41 (95%CI = 1.15–1.76); *p*-value: <0.001], poor access to care [HR = 1.99 (95%IC = 1.70–2.35); *p*-value: <0.0001], an age between 65 and 75 years old [HR = 1.48 (95%IC = 1.09–2.01); *p*-value: <0.05] or >75 years old [HR = 2.21 (95%CI = 1.62–3.01); *p*-value: <0.0001] and being a resident in a non-urban area [HR = 1.35 (95%CI = 1. 08–1.69); *p*-value: <0.01] were all risk factors associated with lower survival estimates. 

Furthermore, the model showed a significant interaction between the municipality of residence and the deprivation index (DC.1 > 0) [HR = 0.69 (95%CI = 0.50–0.96); *p*-value: <0.05], suggesting that the deprivation level may act as a risk factor more in the urban than in the non-urban area.

## 4. Discussion

We have conducted a retrospective observational study to investigate the main demographic factors and socioeconomic conditions potentially associated with HCC patients’ access to healthcare services and HCC survival, in an area with a documented high hepatitis virus infection prevalence and characterised by a changing epidemiological scenario. More in depth, we aimed to estimate whether deprivation, sex, age and access to treatment were factors associated with access to care and HCC survival, during 12-year epidemiological surveillance provided by a population-based cancer registry covering an area of southern Italy with a high prevalence of HCV and the presence of three highly specialised centres for the treatment of HCC patients [18]. To these ends, we innovatively calculated an aggregated weighted deprivation index, obtained as the linear combination of four deprivation variables (people with low education levels, unemployed people, living in rented places, living in crowded houses).

In our study, we documented that HCC occurred more frequently in males and in patients over 65 years old, as previously reported by the literature [36]. Most of the HCC patients were residents in the urban area, therefore having easier access to care probably due to the availability of specialised healthcare facilities, while patients with the poorest access to care lived more frequently in non-urban municipalities where the lack of experienced medical professionals may require travelling long distances to receive an early diagnosis and appropriate treatment [16].

Furthermore, about one-third of HCC cases had limited access to cancer care, and females had the poorest access to healthcare services. Differently, primary liver cancer patients with easy access to care and a morphologically established tumour were males with a younger age than the ones distant from healthcare services. These findings were confirmed by the logistic regression analysis, highlighting how HCC patients living in the non-urban areas, being older than 75 years and female and showing a high level of deprivation had limited access to cancer care.

Social factors may play a fundamental role in determining access to health services, even under conditions of uniform clinical or perceived need, resulting in marked variations in their rates of utilisation [37]. Of interest, pathways of care can be classified into desirable pathways to diagnosis, in which cancers are detected by screening or a specialist with a referral within a 2-week window, and less desirable pathways to diagnosis, which include emergency department presentations and DCO [38]. Further, geographic proximity is a well-known function of distance and travel time, and it is one of the most important aspects of health service accessibility [39]. Although available studies have shown that a decrease in risk for less desirable diagnostic pathways was due to the age and socioeconomic conditions of the patients, this intrinsic relation between socioeconomic determinants and access to care has made the identification of pure risk factor effects challenging [12,40,41]. Moreover, despite the previous evidence that suggested that lower socioeconomic status was associated with an approximately twofold increase in the incidence of HCC [42], conflicting results regarding the association between deprivation and survival after HCC diagnosis have been reported to date [12,43]. In the desire to overcome the contrasting evidence, our study investigated the role of access to care on HCC survival therefore allowing us to obtain further insights into the relationship between socioeconomic factors, geographic location, access to care and hepatocellular carcinoma survival. Specifically, we created two groups of patients with easy and poor access to care, balanced by deprivation components, age, sex and the place of residence, and we estimated their survival before and after the propensity score matching procedure. More in depth, after propensity score matching, we observed a stronger effect of access to care, whereas patients with poor access to care had significantly lower survival. Also, the deprivation component taking into account all the conventional deprivation aspects besides living in a single-parent family resulted in a potential risk factor for HCC survival. This is in line with previous studies on socioeconomic status and HCC survival [44,45].

Moreover, we looked at the interaction between high deprivation levels and residence in a non-urban area, finding out that it may have significant potential implications for the survival rates of HCC patients in our dataset. The data clearly indicated that deprivation, as measured by socioeconomic status, might have had a distinct and negative impact on the survival of HCC patients residing in the city of Palermo. This result potentially implies that patients living within the urban area might be more negatively affected by the role of deprivation. It is important to note that the metropolitan area of Palermo is not only a densely populated urban area but it also houses specialised medical centres renowned for their excellence in HCC treatment. Therefore, the negative influence of deprivation on survival might have been amplified within the city due to the stark contrast between affluent and deprived neighbourhoods.

However, the scenario changed when considering HCC patients residing in the non-urban areas, whereas high levels of deprivation did not seem to have a significant negative impact on their survival rates. Therefore, a plausible explanation for this phenomenon could be the geographic distance from the urban area characterised by more structured healthcare services. Patients living in non-urban municipalities, which often encompass more remote and rural areas, might not experience the same detrimental effects of deprivation as their urban counterparts, as their primary risk factor associated with lower survival appeared to be the geographical remoteness from the metropolitan area. This geographical factor can be seen as a double-edged sword. On one hand, it may limit access to specialised medical centres located in the urban area, potentially leading to delayed or less comprehensive treatment for HCC patients in the non-urban area. On the other hand, the lower population density and potentially reduced environmental risks in provincial areas could mitigate some of the negative impacts associated with deprivation.

In our study, gender and age instead seemed to not play a fundamental role in HCC survival. Gender had no significant effect both before and after the propensity score matching procedure, while age lost its relevance after matching, as demonstrated by the evidence of significantly poorer survival just for patients in the oldest age category.

Although the findings of this observational study appear to be solid according to the cohort design and the sample analysed, there are some limitations that should be discussed. Despite the use of the propensity score that allowed us to mitigate the selection bias and remove the confounding effect of the observed variables, the heterogeneity that could have arisen from unobserved variables has to be highlighted. Moreover, after the matching procedure, there was a significant reduction in the sample size that could have had an impact on the different numerosity of the two groups of patients characterised by poor and easy access to care. Furthermore, the exclusion from the model of HCC patients with the poorest access to care could have caused an overestimation of survival.

Moreover, we limited the period in study prior to the COVID-19 pandemic according to the documented disruption of the spectrum of cancer care, including delays in diagnoses and treatments, the impact of which for HCC patients should be investigated in dedicated future studies [46,47].

Of interest, differently than what was documented for other tumour sites, when comparing the 5-year net survivals prior to and during the COVID-19 pandemic, there was no significant difference attributable to this event with regard to HCC patients, but this evidence needs to be confirmed by further studies [48].

Lastly, the population-based approach, which is typical for cancer registries, differently from evidence provided by clinical-based registries [43], allowed us to obtain data from a cohort of patients that was representative of the entire population of primary liver cancer cases, including the ones occurring in patients distant from care, therefore potentially providing more reasonable insights into the role of deprivation on survival outcomes. However, our results are limited to HCC and might not be generalisable to other types of cancers. Moreover, our findings should be confirmed by further studies to be conducted on larger samples representative of other Italian areas.

Our findings highlighted the importance of targeted prevention strategies and interventions on HCC high-risk groups from different communities, based on both socioeconomic and geographic factors, with the aim to improve health outcomes.

In Southern Italy, over the past few decades, there has been a notable rise in the prevalence of obesity, which is recognised as a significant risk factor associated with liver cancer [49]. Differently, Northern Italy regions traditionally exhibited higher rates of alcohol consumption, a well-established risk factor for HCC, primarily through its direct toxic effects on the liver and steatosis [15], as compared to the southern ones. Therefore, prevention plans and community-based surveillance programs are expected to implement targeted strategies [50,51], based on the risk stratification by the recalled determinants, to reduce social and health disparities and promote the adherence to healthy lifestyles to mitigate alcohol and unhealthy food consumption, according to the epidemiological changing scenario related to primary liver cancer.

In this context, cooperation between the Sicilian population-based cancer registries and the Sicilian clinical Network for Therapy, Epidemiology and Screening in Hepatology (SINTESI), including the primary care and hospital levels [52], has recently been implemented in the framework of the DigitAl lifelong pRevEntion (DARE) initiative, with the aim to provide tools for risk assessment and targeted interventions based on lifestyles, health determinants and environmental profiling and to deploy an advanced interoperability system for integrated epidemiological surveillance and community-based prevention interventions for cancer [53].

## 5. Conclusions

In conclusion, our study provided some new insights into the potential relationship between deprivation, geographic location, access to care and HCC survival. While deprivation appeared to be a significant risk factor for HCC patients residing within the urban area, the geographic distance from specialised treatment centres emerged as a potential determinant of survival estimates for residents in non-urban areas.

Comprehensive national and regional health and prevention plans should take into account the evidence provided to ensure equitable access to HBV vaccination and the early detection of HCV infection, to be monitored through dedicated screening and linkage to care [50,51,54].

Last but not least, to improve the knowledge on the impact of HCC-related emerging risk factors and the effectiveness of the pathways of care leading to liver cancer diagnosis, the pivotal role of the interoperability between population-based cancer registries and other data sources related to health determinants, such as socioeconomic conditions and lifestyle profiling, should be highlighted [55]. In the same direction, the deployment of innovative digital approaches to improve cancer prevention and care should be promoted [55,56].

## Figures and Tables

**Figure 1 cancers-16-02046-f001:**
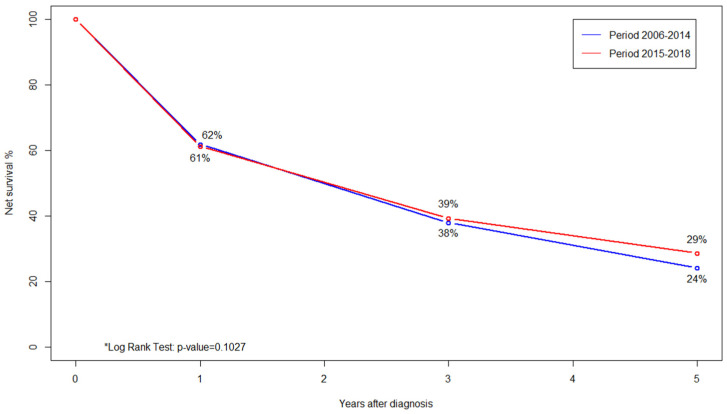
A comparison of 5-year net survival curves in HCC patients diagnosed in the periods 2006–2014 (blue line) and 2015–2018 (red line); the latter intercepts the COVID-19 pandemic. Palermo Province Cancer Registry.

**Figure 2 cancers-16-02046-f002:**
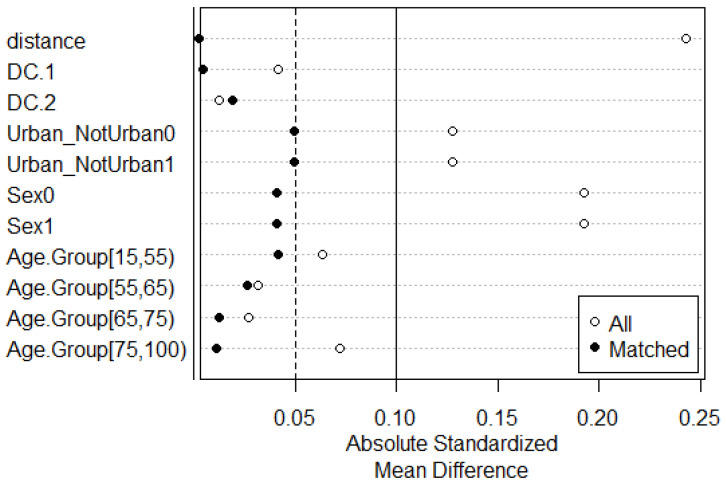
A comparison of absolute standardised mean difference values of the covariates (DC.1, DC.2, the place of residence, gender and age group) in the two balanced groups of patients (“poor” versus “easy access to care”), before (white dots) and after (black dots) applying the nearest neighbour matching of the propensity scores. HCC cases from the Palermo Province Cancer Registry, 2006–2019.

**Table 1 cancers-16-02046-t001:** The distribution of the characteristics of the 2687 patients with a primary liver tumour (C22.0) in the study by access to care. Palermo Province Cancer Registry, 2006–2019.

Characteristics	Total (%)	Access to Care			
		Easy (%)	Poor (%)	Poorest * (%)	*p*-Value
	2687 (100)	1744 (64.9)	700 (26.1)	243 (9.0)	
Gender					
Male (%)	1754 (65.3%)	1213 (69.6)	421 (60.0)	120 (49.4)	<0.00001
Females (%)	933 (34.7%)	531 (30.4)	279 (40.0)	123 (50.6)	
Age at diagnosis (in years)					
Mean	71	69	72	79	<0.0001
15–44 (%)	43 (1.6%)	33 (2.0)	6 (0.9)	4 (1.7)	<0.00001
45–54 (%)	167 (6.2%)	128 (7.3)	36 (5.1)	3 (1.2)	
55–64 (%)	446 (16.6%)	321 (18.4)	107 (15.3)	18 (7.4)	
65–74 (%)	953 (35.5%)	687 (39.3)	228 (36.2)	38 (15.6)	
≥75 (%)	1078 (40.1%)	575 (33.0)	323 (46.1)	180 (74.1)	
Residence					
Urban area (%)	1676 (62%)	1142 (65.5)	415 (59.0)	119 (49.0)	<0.0001
Non-urban area (%)	1011 (38%)	602 (34.5)	285 (41.0)	124 (51.0)	

* Identified by death certificate only (DCO).

**Table 2 cancers-16-02046-t002:** A logistic regression model for the probability of HCC patients to have the poorest access to care, by deprivation components, age groups, sex and residence. Palermo Province Cancer Registry, 2006–2019.

Variables	OR	95%CIs	*p*-Value
Deprivation component 1 (DC.1 > 0)	**1.41**	**1.07–1.87**	**<0.05**
Deprivation component 2 (DC.2 > 0)	1.12	0.84–1.51	0.41
Gender (female)	**1.68**	**1.27–2.22**	**<0.0001**
Age group (55–64)	1.17	0.50–3.07	0.72
Age group (65–74)	1.11	0.52–2.76	0.79
Age group (≥75)	**5.08**	**2.51–12.16**	**<0.0001**
Residence (Urban versus Non-urban area)	**1.97**	**1.45–2.67**	**<0.0001**

OR: Odds Ratio; 95%CIs: 95% Confidence Intervals.

**Table 3 cancers-16-02046-t003:** Fixed effects regression model on the balanced dataset used to estimate net survival in HCC patients by deprivation indexes, access to care, age and municipality of residence. Palermo Province Cancer Registry, 2006–2019.

Variables	Before Matching			After Matching		
	HR	95%CIs	*p*-Value	HR	95%CIs	*p*-Value
Deprivation component 1 (DC.1 > 0)	**1.29**	**1.12–1.48**	**<0.001**	**1.38**	**1.12–1.70**	0.002
Deprivation component 2 (DC.2 > 0)	1.07	0.96–1.20	0.22	0.90	0.76–1.06	0.20
Access to care (poor to care)	**2.08**	**1.84–2.36**	**<0.0001**	**2.25**	**1.91–2.66**	**<0.0001**
Gender (female)	1.03	0.93–1.15	0.57	0.96	0.82–1.11	0.56
Age group (65–75)	**1.29**	**1.06–1.57**	**<0.05**	1.17	0.87–1.58	0.29
Age group (≥75)	**1.83**	**1.50–2.23**	**<0.0001**	**1.79**	**1.33–2.42**	**<0.0001**
Municipality of residence (Non-urban area)	**1.32**	**1.13–1.54**	**<0.001**	**1.47**	**1.17–1.84**	**0.0008**
Interaction between DC.1 > 0 and residence in a Non-urban area	0.81	0.65–1.01	0.06	**0.69**	**0.50–0.95**	**0.02**

HR: hazard ratio; 95%CIs: 95% Confidence Intervals.

## Data Availability

The data analysed in the current study are available from the corresponding author on reasonable request with the permission of Palermo Province Cancer Registry. However, restrictions apply to the availability of this data, which were used under license for the current study.

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
