# Peer review of "Factors Associated with Primary Liver Cancer Survival in a Southern Italian Setting in a Changing Epidemiological Scenario"

_cancers, 2024, doi:10.3390/cancers16112046_

Round 1

Reviewer 1 Report

Comments and Suggestions for Authors

A retrospective observational study with cancer incidence data from a population-based registry. Poor access to care and a lower socioeconomic status were negatively associated with a  primary liver cancer survival, particularly among urban residents.

Major issues

1.      Authors used retrospective study design. From epidemiological point of view, retrospective studies do not allow conclusions about causal relationships but only statistical associations. Authors however, on various places of the manuscript write sentences meaning causal relationships when they write “is a risk factor,” “resulted to negatively impact,” “uncovered a nuanced relationship.”     Only in the first part sentences of the discussion, they correctly write about associations. I recommend rephrasing all sentences in discussion where they conclude such causal effects and rather write about associations.

2.      Authors should add more relevant imitations in discussion , for example that these findings maybe cannot be extrapolated to other regions or countries, and that they used complex statistical methodology which- like each complex biometrical method- may enables very bride interpretations of results. Also, that the study is limited to only one relative rare cancer type.

Minor issues

1.      Kaplan-Meier curves would be good to see, not only because they usually look and clear but also because they are proof that differences between survival rates are similar over time.

Author Response

Major issues

  1. Authors used retrospective study design. From epidemiological point of view, retrospective studies do not allow conclusions about causal relationships but only statistical associations. Authors however, on various places of the manuscript write sentences meaning causal relationships when they write “is a risk factor,” “resulted to negatively impact,” “uncovered a nuanced relationship.”     Only in the first part sentences of the discussion, they correctly write about associations. I recommend rephrasing all sentences in discussion where they conclude such causal effects and rather write about associations.

Reply: 

Authors sincerely thank the reviewer for the comment. We have reviewed and rephrased the entire text of the manuscript as suggested (please see the track changes).

  1. Authors should add more relevant limitations in discussion, for example that these findings maybe cannot be extrapolated to other regions or countries, and that they used complex statistical methodology which- like each complex biometrical method- may enables very bride interpretations of results. Also, that the study is limited to only one relative rare cancer type.

Reply:

Authors thank again the reviewer for the comment. We have reviewed and rephrased the text of the manuscript as suggested, indicating the limit of extrapolation to other regions or countries and to the specific type of cancer. Anyway, despite the complexity of the statistical methodology, we are still confident that our model could be extended to other areas. We have implemented the discussion as follows (Page 9, lines 360-361): “However, our results are limited to HCC and might not be generalized to other type of cancers. Moreover, our findings should be confirmed by further studies to be conducted on larger samples representative of other Italian areas.

Minor issues

  1. Kaplan-Meier curves would be good to see, not only because they usually look and clear but also because they are proof that differences between survival rates are similar over time.

Reply

Many thanks for the comment. We agree with the reviewer, anyway, we would prefer not to insert any additional figure, which could overload the manuscript.

Reviewer 2 Report

Comments and Suggestions for Authors

Very interesting and nicely written study. Like all these regional/national database studies, the analysis lacks of granularity and several baseline informations are missing. However, it represents a nice addition to the body of evidence on HCC management.

The classification in easy to access care doesn't seem really validated and it should be supported better by the current literature

The concept of deprivation is well known in the literature....can the authors confirm their approach is in line with other similar papers in other fields of oncology? 

The topic seems similar to a recent ITALICA paper PMID: 34666215.....which are the main differences?

Author Response

1. The classification in easy to access care doesn't seem really validated and it should be supported better by the current literature

Reply:

Authors thank the reviewer for the comment. As stated in the methods section, we used a linkage algorithm, previously validated by our research team, to stratify primary liver cancer patients by access to care.  In brief, while assuming that a higher or lower number of contacts with the healthcare service could be considered a proxy for an easier or poorer access to healthcare, we explored access to care, using different data flows: hospital discharge records (HDRs), ambulatory discharge records (ADRs), Pathological Anatomy Reports (PARs), death certificate (DCO). Patients linking with at least two different healthcare sources, including PARs, were considered to have an “easy access to care”, while patients linking with only a HDR or ADR, but no PAR, were defined as “poor access to care” ones. Patients diagnosed only through a DCO were excluded from survival analysis. To make it clearer to the reader, we have implemented the methods section as follows (Pages 2 and 3, lines 102-108): “We categorised primary liver cancer patients by access to care according to a previously validated linkage algorithm that allowed us to identify patients with an “easy access to care”, the ones having a higher number of contacts with the healthcare service as confirmed by the linkage with at least two different healthcare sources (hospital discharge records, ambulatory discharge records, and/or pathological anatomy reports), and patients having a “poor access to care”, that were identified by a single data source [7].

2. The concept of deprivation is well known in the literature.... can the authors confirm their approach is in line with other similar papers in other fields of oncology?

Reply:

Authors sincerely thank the reviewer for the comment. We have clearly stated in the discussion that our findings on the relationship between deprivation and HCC survival are in line with the literature, though we further investigated the role of the interaction between deprivation and living in a non-urban area, that as far as author knowledge has not been investigated in the literature. Please see Page 8, lines 294-295: “This is in line with previous studies on socioeconomic status and HCC survival [45,46].

3. The topic seems similar to a recent ITALICA paper PMID: 34666215.....which are the main differences?

Again, we thank the reviewer for the comment. As well as we know, the ITA.LI.CA. consortium provides real-world evidence using data from a clinical-based registry, which contains detailed individual and clinical information on a cohort of patients with hepatocellular carcinoma under clinical surveillance. More in depth, the manuscript mentioned by the reviewer aimed to explore how material deprivation in Italy might have influenced the stage of hepatocellular carcinoma (HCC) at diagnosis and, therefore, the chance of cure and the survival outcomes. However, this cohort of patients may suffer from a selection bias related to the capacity of the clinical network to intercept patients distant from care. That’s why the proposed population-based approach, which characterises the cancer registries, allows to obtain data from huge cohorts of patients which are representative of the entire population of primary liver cancer cases, including the ones distant from care, therefore providing more affordable information on the role of deprivation. As we believe that both the clinical- and the population-based approaches are of interests for the scientific community, we have implemented the text to highlight the different points of view (Pages 8 and 9, lines 342-359): “Lastly, the population-based approach, which is typical for cancer registries, differently from evidence provided by clinical-based registries [48], allowed us to obtain data from a cohort of patients which was representative of the entire population of primary liver cancer cases, including the ones occurring in patients distant from care, therefore potentially providing more affordable insights with regard to the role of deprivation on the survival outcomes”

Reviewer 3 Report

Comments and Suggestions for Authors

The manuscript "Factors associated with primary liver cancer survival in a Southern Italian setting in a changing epidemiological scenario" is an outstanding study of the incredible situation in this country. The disparity in health accessibility is analyzed in depth by the authors.

A dramatic consideration by the authors, "about one-third of HCC cases had limited access to cancer care, and women had the poorest access to health services." If we consider these results in an EU state.

National and regional health and prevention plans needed to decrease this prevention disparity in Southern Italy are out of the question.

This study should be sent to politicians and health officials for timely action. 

Author Response

The manuscript "Factors associated with primary liver cancer survival in a Southern Italian setting in a changing epidemiological scenario" is an outstanding study of the incredible situation in this country. The disparity in health accessibility is analyzed in depth by the authors. A dramatic consideration by the authors, "about one-third of HCC cases had limited access to cancer care, and women had the poorest access to health services." If we consider these results in an EU state.National and regional health and prevention plans needed to decrease this prevention disparity in Southern Italy are out of the question.This study should be sent to politicians and health officials for timely action.

Reply:

We sincerely thank the reviewer for the appreciation given to our manuscript. Following our experience, we are working to extend the model to the entire Sicilian region, and we are in contact with the Regional Health Authority to use the evidence to define targeted preventive interventions.

Round 2

Reviewer 1 Report

Comments and Suggestions for Authors

Authors addressed my comments and wrote that they do want include Kaplan-Meier curves into manuscript. On the one side, this is ok; on the other side, the reviewer is still not sure about the hazard assumption when K-M curves are  not shown. The solution I suggest is not to include curves in manuscript but show them in the reply to the reviewer.

Reviewer 2 Report

Comments and Suggestions for Authors

The revised version of the manuscript is OK. Thank you!

Author Response

Authors thank the reviewer.